# An approach to quantitate maternal transcripts localized in sea urchin egg cortex using RT-qPCR with accurate normalization

**Yulia O. Kipryushina, Mariia A. Maiorova, Konstantin V. Yakovlev**<sub></sub> *

Laboratory of Cytotechnology, A.V. Zhirmunsky National Scientific Center of Marine Biology, Far Eastern Branch, Russian Academy of Sciences, Vladivostok, Russia

* konstantin.yakov@gmail.com

## Abstract

The sea urchin egg cortex is a peripheral region of eggs comprising a cell membrane and adjacent cytoplasm, which contains actin and tubulin cytoskeleton, cortical granules and some proteins required for early development. Method for isolation of cortices from sea urchin eggs and early embryos was developed in 1970s. Since then, this method has been reliable tool to study protein localization and cytoskeletal organization in cortex of unfertilized eggs and embryos during first cleavages. This study was aimed to estimate the reliability of RT-qPCR to analyze levels of maternal transcripts that are localized in egg cortex. Firstly, we selected seven potential reference genes, 28S, *Cycb*, *Ebr1*, *GAPDH*, *Hmg1*, *Smtnl1* and *Ubb*, the transcripts of which are maternally deposited in sea urchin eggs. The candidate reference genes were ranked by five different algorithms (BestKeeper, CV, ΔCt, geNorm and NormFinder) based on calculated level of stability in both eggs as well as isolated cortices. Our results showed that gene ranking differs in total RNA and mRNA samples, though *Ubb* is most suitable reference gene in both cases. To validate feasibility of comparative analysis of eggs and isolated egg cortices, we selected *Daglb-2* as a gene of interest, which transcripts are potentially localized in cortex according to transcriptome analysis, and observed increased level of *Daglb*-2 in egg cortices by RT-qPCR. This suggests that proposed RNA isolation method with subsequent quantitative RT-qPCR analysis can be used to determine cortical association of transcripts in sea urchin eggs.

## Introduction

RT-qPCR is a powerful tool to quantify gene expression levels during development, after exposure to chemical or physical treatment of cells in *in vitro*. Data normalization in RT-qPCR analysis is aimed to minimize errors in estimation of target mRNA levels. The most common approach is usage of endogenous reference genes [1]. Perfect reference gene should have constant expression, while expression levels of many genes may be considerably changed during development and reveal different expression levels in different tissues and organs. So, each particular case requires seeking for reference genes that are most stably transcribed in all

**Data Availability Statement:** SRAs and assembled transcriptome are available in GenBank (BioProject PRJNA686841).

**Funding:** This work is financially supported by the Russian Foundation for Basic Research grant 20-04-00332 (K.V.Y.). The funders had no role in study design, data collection and analysis, decision to publish, or preparation of the manuscript.

**Competing interests:** There are no Competing Interests to declare.

experimental samples. Using variably expressed genes as references leads to incorrect results [2]. Reference genes are chosen by comprehensive evaluation of gene expression stability of candidate genes by combinations of several methods.

Egg is a single cell, which has the potential to develop into a multicellular organism. Oocytes and eggs are polarized by asymmetrically deposited organelles and molecules within the cytoplasm. Asymmetrically distributed maternal molecules, RNAs and proteins, are key regulators of cell specification during early development. One of the ancient mechanisms governing cell polarization is associated with localized RNAs found in oocytes of many model animals, like ascidians, *Drosophila*, zebrafish and *Xenopus*. Localized RNAs are also found in somatic cells, like neurons, oligodendrocytes, myoblasts, fibroblasts and epithelial cells [3,4]. In oocytes, different types of cytoskeleton play a major role in anchoring of RNAs [5]. Drosophila *nanos* is accumulated by diffusion and entrapment posteriorly by binding to actin filaments [6]. Localization of *gurken* requires static anchoring by Dynein at dorsal-anterior oocyte region and *oskar* posterior accumulation depends on its interaction with Kinesin heavy chain [7,8]. In *Xenopus* oocytes, *Vg1* RNA is actively transported along microtubes and anchored to actin microfilaments in vegetal oocyte cortex [9].

In sea urchin eggs, cortex may play a key role for accumulation of maternal factors that lead to establishment of polarity along both animal-vegetal and dorsal-ventral axes [10–12]. Disheveled, a protein of the Wnt/β-cathenin pathway regulating specification of vegetal blastomeres, is found in vegetal part of the eggs joined with egg cortex [13]. Also, *Panda* and *Coup-TF* mRNAs are found in subcortical area of oocytes, unfertilized eggs and early embryos. *Panda* reveals gradient distribution is required to restrict Nodal signaling, which leads to dorsal-ventral axis formation in the sea urchin embryos [14]. Coup-TF is a member of steroid-thyroid-retinoic acid superfamily, which controls proper cell specialization along both animal-vegetal and dorsal-ventral axes. *Coup-TF* knockdown leads to lack of nervous and digestive systems and ciliary band in embryos [15]. Unequal distribution of maternal *Coup-TF* mRNA has been detected but not in all sea urchin species. *Coup-TF* were found to be localized laterally to animal-vegetal and 45° angle to dorsal-ventral axes in eggs of *Strongylocentrotus purpuratus* and *Lytechinus variegatus*, but not of *Paracentrotus redivivus* [16,17]. Some proteins necessary for development are associated with egg cortex, suggesting cortical distribution irrespective of directions of prospective developmental axes. Seawi and Vasa have been found in granules localized in egg cortex and later accumulated in primordial germ cells of sea urchin embryos [18,19]. Besides specified animal-vegetal and dorsal-ventral axes in sea urchin eggs early segregation of apical and basolateral cortical regions with involvement of Par proteins [20] suggest the presence of other localized maternal factors that are necessary for epithelial organization of blastoderm.

Cell specification along embryonic axes and establishment of architecture of embryonic cells require many unequally distributed maternal factors in oocytes and eggs, many of them are still unknown for sea urchins. Exciting approach for quantitative RNA measurement called qPCR tomography was designed on *Xenopus* oocytes [21,22]. Principles of this method consist of RT-qPCR with RNA samples isolated from cryosections of oocytes given along animal-vegetal axis. Thus, the authors propose to use qPCR tomography to analyze spatial expression patterns of RNAs in *Xenopus* oocytes localized in animal and vegetal poles. Availability of appropriate methods is a good prerequisite for further development of methods to study spatial distribution of maternal transcripts in sea urchin oocytes and eggs. One of the perspective approaches is comparative quantitative analysis of RNAs from sea urchin eggs and their isolated cortices. A method for cortex isolation from sea urchin eggs and embryos since 1970s [23]. In the current study, we used this method to isolate RNA with following RT-qPCR, which allows measurement of levels of cortex-associated maternal transcripts.

The primary goals of this study are to evaluate suitability of quantitative RT-qPCR analysis of egg cortex-associated maternal transcripts and find appropriate reference genes for accurate signal normalization. Firstly, we found that RNA isolation from egg cortices is a feasible procedure with additional stages to further concentrate RNA samples. We selected some previously known (28S, *GAPDH*, *Hmg1* and *Ubb*), and several new (*Cycb*, *Ebr1* and *Smtnl1*) candidate reference genes and performed RT-qPCR analysis of egg and isolated egg cortices. This set of genes was subjected to expression stability analysis using BestKeeper [24], coefficient of variation (CV) [25], ΔCt [26], geNorm [27] and NormFinder [28] methods. We found that stability of most selected genes diverged in total RNA and poly(A) RNA samples. The highest stability in both cases was for *Ubb*, which encodes polyubiquitin. So, this is the most suitable reference gene for comparative analysis of egg and cortex samples. Further, we predicted cortical localization of *Daglb-2* by transcriptome analysis and analyzed its levels in the eggs and cortices using RT-qPCR. We compared expression levels in both total and mRNA samples and found higher level of *Daglb-2* in mRNA samples of isolated cortices. This finding suggested that usage of mRNA fractions is effective in determining cortical association of *Daglb-2* in sea urchin eggs. Our results demonstrate a possibility to perform RT-qPCR analysis of isolated sea urchin egg cortices with accurate signal intensity normalization.

## Materials and methods

### Animals and sample preparations

Adult *S. intermedius* sea urchins were collected in the Peter the Great Bay (Sea of Japan) (permission to collect animals 252021030802 of the Federal Agency for Fishery of the Russian Federation), kept in tubes with aerated sea water and fed with algae (*Ulva fenestrata* and *Saccharina japonica*) and carrot. Eggs were obtained by injection with 0.5M KCl. Eggs were washed several times with filtered sea water and then two times with CFSS (12mM HEPES, pH 7.6–7.8, 385mM NaCl, 10mM KCl, 21mM $Na_2SO_4$, 17mM glucose and 2.5mM $MgCl_2$). Cortices were isolated as described previously [19,29–31]. Briefly, eggs attached to poly-L-lysine-coated coverslips (24×24 mm) were washed twice with CFSS supplemented with 5 mM EGTA. The coverslips were then gently washed by direct sprinkling with cortex isolation buffer (0.8 M mannitol, 50 mM Hepes, 50 mM Pipes, pH 6.5–6.8, 2.5 mM $MgCl_2$, 20 mM EGTA, titrated by KOH) to remove majority of the egg content. Cortex samples were immediately used for RNA isolation. Isolated cortices were prepared on 6–8 coverslips for RNA isolation. To confirm quality of the isolated cortices, the latter were processed for imaging. The cortices were fixed with 3% PFA and 0.1% glutaraldehyde in CIB for 30 min. Coverslips were washed with PBS, mounted in Vectashield and observed using phase contrast and DIC microscopy on Axio Imager A2 equipped with AxioCam HRc digital camera (Carl Zeiss, Germany). For actin labeling, after fixation and washing, the cortices were treated with 0.1 M glycine in PBS (15 min) and then blocked with 1% BSA (20 min). Cortices were stained with phalloidin-CruzFluor 488 (1:150, Santa Cruz, USA) for 1 h, washed and mounted in Vectashield (Vector Laboratories, USA). Confocal images were taken on LSM 710 LIVE (Carl Zeiss, Germany). Images were processed using Fiji software [32].

### RNA extraction and cDNA synthesis

Total RNA was extracted from unfertilized eggs and isolated egg cortices using PureLink Mini kit (Thermo Fisher Scientific, USA) with some modifications. Total RNA from eggs was isolated according manufacturer's manual from 4–5 µl of egg suspension using 0.6 ml of lysis buffer supplemented with DTT. RNA from cortices were isolated using 2.5 ml of lysis buffer per 6–8 coverslips. Each coverslip was consequently placed in Petri dish filled with lysis buffer

and cortices were lysed by pipetting. Then, the content was centrifuged through one spin cartridge after addition of equal volume of ethanol. To analyze total RNA, the samples were subsequently concentrated by GeneJet RNA Cleanup and Concentration Micro kit (Thermo Fisher Scientific, USA). Concentrations of the samples were estimated by measuring absorbance at 260 nm on Biophotometer (Eppendorf, Germany). Only samples with high purity (A260/A280 = 1.8–2.0) were used in analysis. To obtain poly(A) mRNA fraction, the RNA samples isolated by PureLink Mini kit were subsequently purified by Magnetic mRNA Isolation kit (New England Biolabs, USA). mRNA concentration was estimated by Qubit RNA HS Assay Kit (Thermo Fisher Scientific, USA). The first strand cDNA was synthetized using ProtoScript II kit (New England Biolabs, USA) from 1 μg of total RNA or 1.5 ng of mRNA with Random Primer Mix (2 and 0.5 μl, respectively). cDNA samples were diluted two times and stored at -80˚C until further use.

## Transcriptomic analysis

RNA-sequencing was done on entire eggs and isolated cortices. Illumina TruSeq stranded mRNA library construction and generation of raw sequence reads using Illumina NovaSeq 6000 platform (2×100 pair bases) were performed by Macrogen Company (Seoul, South Korea). *De novo* transcriptome assembly was built by Trinity [33] using Galaxy web-based platform [34] (https://usegalaxy.org). SRAs and assembled transcriptome were submitted to GenBank (BioProject PRJNA686841). The assembled sequences were blasted against Uniprot Swiss-Prot database and against *S. purpuratus* genome [35] with a cut-off E-value of 1e-5. For quantitative gene expression analysis reads were aligned to the assembled transcriptome with Bowtie [36], and transcript abundance was estimated with RSEM [37]. We analyzed the gene expression values presented in FPKM. Blast and subsequent analysis were performed using computational resources provided by the Shared Services Center "Data Center of FEB RAS" (Khabarovsk) [38].

## Selection of candidate reference genes and gene of interest

28S rRNA gene and three protein-coding genes, *GAPDH*, *Hmg1* and *Ubb*, were previously used as reference genes for embryonic and adult samples of different sea urchin species [39–42]. Three protein-coding genes, *Cycb*, *Ebr1* and *Smtnl1*, were selected from a list of genes upon preliminary differential expression analysis [37]. *Cycb*, *Ebr1* and *Smtnl1* transcripts are abundant and their FPKM values did not significantly differ in samples of eggs and isolated cortices (Table 1). Gene of interest, *Daglb-2*, was selected from a list of cortically enriched transcripts with FPKM values >0.5 and significantly higher in cortices (Table 1). All used protein-coding sequences were found in the transcriptome. A part of 28S sequence was amplified and sequenced with primers designed to close species *S. purpuratus* (GenBank Ac. No. AF212171.1): Forward (CGCCCAACAGCTGACTCAGA) and Reverse

**Table 1.  FPKM values of new candidate reference genes and gene of interest.**

| Gene | FPKM | |
|---|---|---|
| | **Eggs** | **Cortices** |
| *Cycb* | 18086.99 | 15769.9 |
| *Daglb-2* | 2 | 9 |
| *Ebr1* | 7476.33 | 6490.31 |
| *Smtnl1* | 4183.87 | 3781.28 |

(TAGCACCAGAAATCGGACGAA). All sequences were deposited in GenBank database. Accession numbers of sequences, primers and products' sizes are given in Table 2.

## RT-qPCR

RT-qPCR was conducted on CFX96 Touch Real-Time PCR Detection System (Bio-Rad, USA) using qPCRmix-HS SYBR master mix (Evrogen, Russia). Reaction mixture (25 μl) contained 2 μl of template cDNA and 0.25 μM of each primer with the following temperature program: 94˚C for 30 s, 40 cycles of 94˚C for 10 s, 55˚C for 25 s and 72˚C for 15 s. After then, melting curve analysis was done. Three independent biological replicates were prepared and each replicate was analyzed in technical triplicate. PCR efficiency was evaluated using the CFX Manager (Bio-Rad, USA).

## Data analysis

The stability of seven potential reference genes were analyzed using five approaches, Best-Keeper [24], CV [25], ΔCt [26], geNorm [27] and NormFinder [28]. *Daglb-2*, which is potentially localized in egg cortex, was used to validate selected reference genes. Relative levels of *Daglb-2* in eggs and egg cortices were calculated according to their Ct values using the $2^{-\Delta\Delta Ct}$ method [43]. Statistical analysis and data visualization were performed using GraphPad Prizm 9 Demo software (GraphPad Software, USA).

## Results

### Quality of isolated cortices and measurement of RNA amount

Only high-quality samples were used for RNA isolation. Whole eggs or highly disrupted cortices were absent on the coverslips (Fig 1A). The presence of cortical granules and specific actin pattern point to integrity of the isolated cortices (Fig 1B and 1C). Yield of total RNA isolated from eggs varied from 6 to 19 μg. Yield of RNA isolated from cortices was 1.2–4 μg. To determine percentage of purified poly(A) RNA, mRNA values were divided by total RNA values given for mRNA purification and then multiplied by 100 in each experiment. Amount of mRNA was 84–300 ng (1.43(mean) ± 0.23 (SD)% of input) from eggs and 8.4–18.7 ng (0.64

**Table 2. Names of genes, primers used for RT-qPCR and reaction efficiency.**

| Gene | Accession number | Primers | Product size, bp | Efficiency, % | $R^2$ |
|---|---|---|---|---|---|
| 28S | MW915850 | F: GATTAACGAGATTCCCACTGTCC<br>R: AAGCACCTCCCACCTATCCTAC | 160 | 94 | 0.997 |
| *Cycb* | MW735848 | F: TCACATCAAACCCATCATCCA<br>R: TGATTTCAGCTGTGAGAGCGA | 139 | 91 | 0.996 |
| *Daglb-2* | OK274215 | F: GTATTAGACCCTCTGAGCGCATC<br>R: CCTCTGATTGCGATGACCACT | 187 | 98.4 | 0.995 |
| *Ebr1* | MW735849 | F: AGAAGTGGGAGTTTTCCTTATCCTC<br>R: ACAGGACAGTCCACTGGGTGAT | 195 | 90.1 | 1.000 |
| *GAPDH* | MW735850 | F: GATCTAACTGTCCGTCTGAAGAAGC<br>R: GGGCGATACCAGCGTTAGC | 181 | 105.2 | 0.991 |
| *Hmg1* | MW735851 | F: ACAGAGCAGCCATAAAGAGTGTTC<br>R: TCCTTAGCAGCACCCTTGTCA | 127 | 101.9 | 0.999 |
| *Smtnl1* | MW735852 | F: CAAGTTTGGTGGAGTGGCG<br>R: GCACTGATACCCGTGGTTGTT | 144 | 92 | 1.000 |
| *Ubb* | MW735853 | F: TTCAAAGGCAAGACCATCACAC<br>R: AGAGAGTGCGGCCATCCTC | 148 | 92.5 | 0.999 |

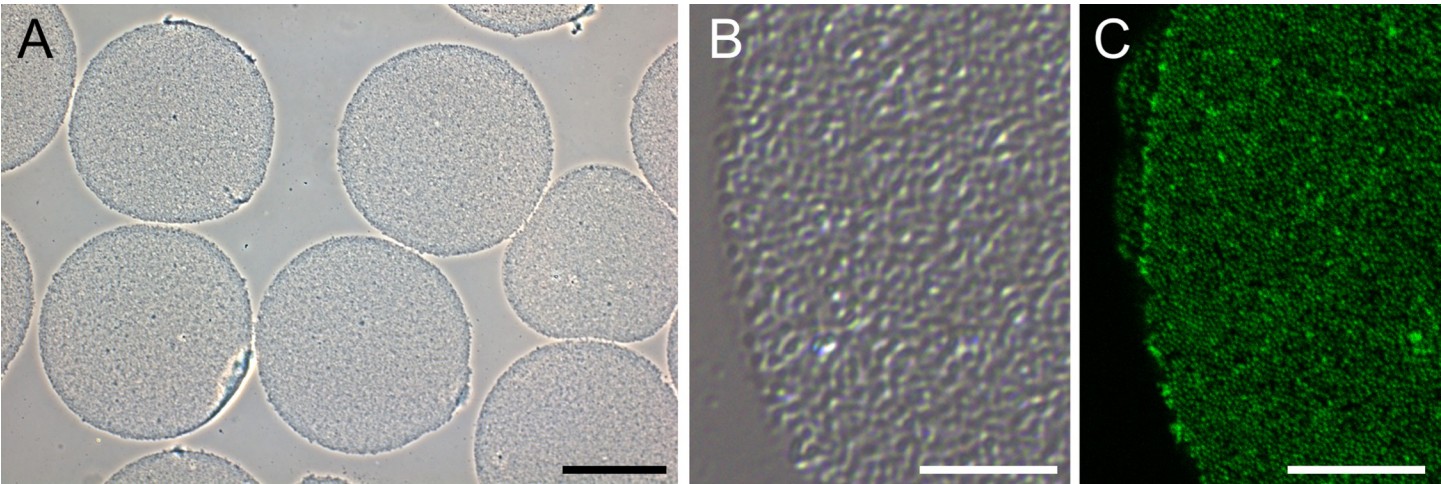

**Fig 1. Isolated cortices from unfertilized eggs.** (A) Phase contrast image of isolated cortices attached to poly-L-lysine treated coverslip. (B) DIC image of cortex at high magnification. Granular pattern indicates multiple cortical granules. (C) Confocal image of actin staining. There is small punctate pattern of actin staining, which is typical for egg cortices. Scale bars 50 μm (A), 10 μm (B, C).

(mean) ± 0.12 (SD)% of input) from cortices. Percentage of purified poly(A) RNA from isolated cortices was 2.27(mean)±0.34(SD) times lower than that from eggs.

## Selection of candidate reference genes, specificity and amplification efficiency of RT-qPCR

Seven potential reference genes for comparative RT-qPCR analysis of eggs and isolated cortical layers were tested to find appropriate genes that can be used for accurate normalization. Four genes were selected based on literature data: 28S, *GAPDH*, *Hmg1*, *Ubb*. Three genes, *Cycb*, *Ebr1* and *Smtnl1*, were selected from a list of preliminary tested genes that are abundant in both eggs and isolated cortices based on transcriptomics analysis (Table 1). Also, we tested several genes previously used for normalization or found by transcriptome analysis, but we omitted them because PCRs specific for these genes using our primers did not fit the required amplification efficiency (90–110%).

All chosen genes were tested for reaction specificity which was determined by melting curve analysis. Single peaks were detected for all tested genes (S1 Fig). No signals were detected with all primer pairs without templates. Amplification efficiencies were calculated by standard curve method using two-, four- or five-fold serial dilutions of cDNA samples. Amplification efficiencies ranged between from 90.1% to 105.2%. Correlation coefficients ($R^2$) displayed values 0.991–1.000 (Table 2).

## Levels of candidate reference genes

Levels of tested candidate reference genes in six samples (three samples from eggs and three samples from isolated egg cortices) were evaluated by RT-qPCR of total RNA and mRNA samples. Raw and mean Ct values are shown in S1 Table. Maximum differences in Ct values between technical replicates were <0.5 cycles. In total RNA samples, among tested genes 28S and *Ubb* were the most abundant genes with the lowest means of Ct values (15.6 and 15.91, respectively). *GAPDH* showed the lowest level with highest mean Ct value (28.02). Maximum and minimum Ct variation were observed for 28S (4.87 cycles) and *Smtnl1* (0.66 cycles), respectively (Fig 2A). In mRNA samples, most abundant genes were *Ubb* (mean Ct value

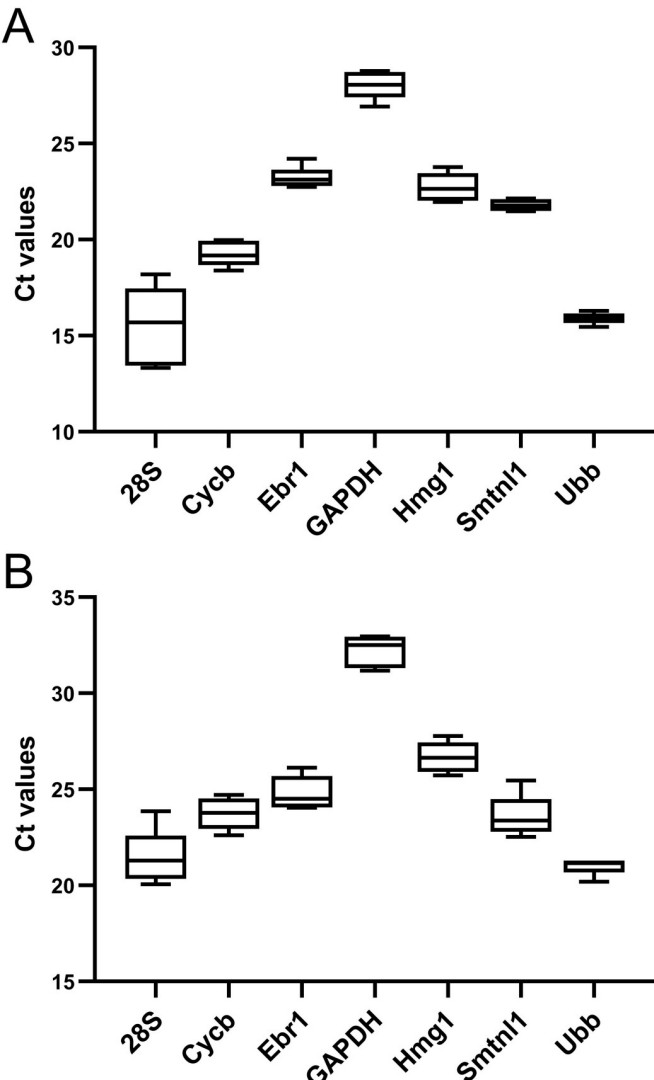

**Fig 2. Boxplot of Ct values for candidate reference genes in all samples.** (A) Ct values in cDNA samples synthesized from total RNA (B) Ct values in cDNA samples synthesized from mRNA. The boxes show interquartile range (25–75%), horizontal lines represent medians. The whiskers show the minimum and maximum values. No outliers were detected.

20.99) and 28S (mean Ct value 21.52). *GAPDH* revealed minimal level with mean Ct of 32.24. 28S was most variative with Ct range of 3.79 cycles. The least variative gene was *Ubb* with Ct range of 1.1 cycles (Fig 2B).

## Expression stability analysis and determination of minimal number of reference genes for normalization

We employed expression stability analysis using five different algorithms to evaluate of level variations for each transcript in unfertilized eggs and egg cortices.

BestKeeper analysis: this method allows to analyze stability by SD and CV generated from raw Ct values [24]. The lowest SD and CV values correspond to the highest stability. SD values ≤1 indicate acceptance as reference genes. According to the SD values, the most stable gene

for total RNA and mRNA samples was *Ubb* (SD value = 0.21 and 0.32, respectively) (Fig 3A). The least stable gene was 28S (SD = 1.72 for total RNA and 0.99 for mRNA). Total RNA value is higher than 1, which is unacceptable for usage of 28S as reference gene. For mRNA, the level of 28S rather reflect the degree of purification.

CV analysis: CV method is simply based on comparison of CV of expression levels. The lowest CV value, which is defined as a ratio of SD to average $2^{Ctmin-Ctsample}$, corresponds to the highest intragroup stability [25]. The least stable gene in both total RNA and mRNA samples was 28S with CV values of 103.4% and 68.17%, respectively. The most stable gene in total RNA samples was *Smtnl1* (CV: 18.41%). In mRNA samples, *Ubb* showed highest stability (CV: 32.43%) (Fig 3B).

ΔCt analysis: the ΔCt method is based on pairwise comparisons and calculation of SD of ΔCt values for each pair of genes [26]. The lowest value of average SD corresponds to the highest stability of expression among evaluated genes. Our results showed that for total RNA the most stable gene was *Ebr1* (SD: 0.66) and the least stable was 28S (SD: 1.75) (Fig 3C). In mRNA samples, the most stable gene was *Hmg1* (SD: 0.743) and least stable gene was again 28S (SD: 1.72).

geNorm analysis: this method is based on pairwise variation that consequently exclude least stable genes after each step of analysis. Finally, two most stable genes are determined. geNorm utilize average expression stability (M) values [27]. Threshold M value of ≤ 0.5 indicate good reference genes. The most stable genes have the lowest M values. As shown, among seven tested genes the best pair for total RNA was *Smtnl1*/*Ubb* with M value of 0.16 (Fig 3D). 28S and *Cycb* showed M values ≥ 0.5, which indicated their inapplicability as reference genes. The best pair in mRNA analysis was *GAPDH*/*Cycb* with M value of 0.33 followed by *Hmg1* (M value 0.4). M values of other genes were above 0.5, which makes them inappropriate as reference genes. Among them, 28S was the least stable (M value 0.94) (Fig 3D). Another parameter calculated by geNorm is pairwise variation ($V_{n/n+1}$) between normalization factors. It allows defining minimal number of reference genes for accurate normalization. Cut-off threshold of 0.15 is recommended to determine the optimal number reference genes [27,44]. In our test, all pairwise variations in both total RNA and mRNA cases revealed M value <0.15 (Fig 4), which point to usage of two reference genes for normalization.

NormFinder analysis: this method takes into account both intragroup and intergroup expression variability. The most stable genes have the lowest stability values [28]. NormFinder analysis revealed that the most and the least stable genes were *Ebr1* (0.125) and 28S (0.97), respectively, in total RNA (Fig 3E). Additionally, NormFinder determined the *Ebr1*/*Hmg1* pair as the best combination of reference genes, as these genes have the lowest values. In mRNA samples, *Cycb* and 28S revealed the highest (0.2) and the lowest stability (0.98), respectively. The best pair of reference genes according to NormFinder was *Cycb*/*GAPDH*.

After analysis by different methods, we summarized the ranking of candidate reference genes, which is presented in Table 3. For total RNA, general view showed that *Ubb*, *Ebr1* and *Smtnl1* were the three most stable, and therefore, the most appropriate reference genes. Best-Keeper analysis revealed that *Ubb* was most stable gene. ΔCt and NormFinder analyses detected *Ebr1* as most stable and according to CV method the most stable gene was *Smtnl1*. geNorm does not allow recognition of the best reference gene, this method determines the pair of genes with the highest stability. For total RNA the best pair of reference genes was found to be *Smtnl1*/*Ubb*. Data obtained from mRNA samples showed different ranking of genes (Table 3). *Ubb* was ranked as most stable by BestKeeper and CV methods. ΔCt determined *Hmg1* as the most stable gene, and *Cycb* was the most appropriate reference gene according to NormFinder analysis. geNorm revealed *Cycb*/*GAPDH* as the most stable pair. Only one gene, *Ubb*, was found among the most stable genes in two separate analyses of total

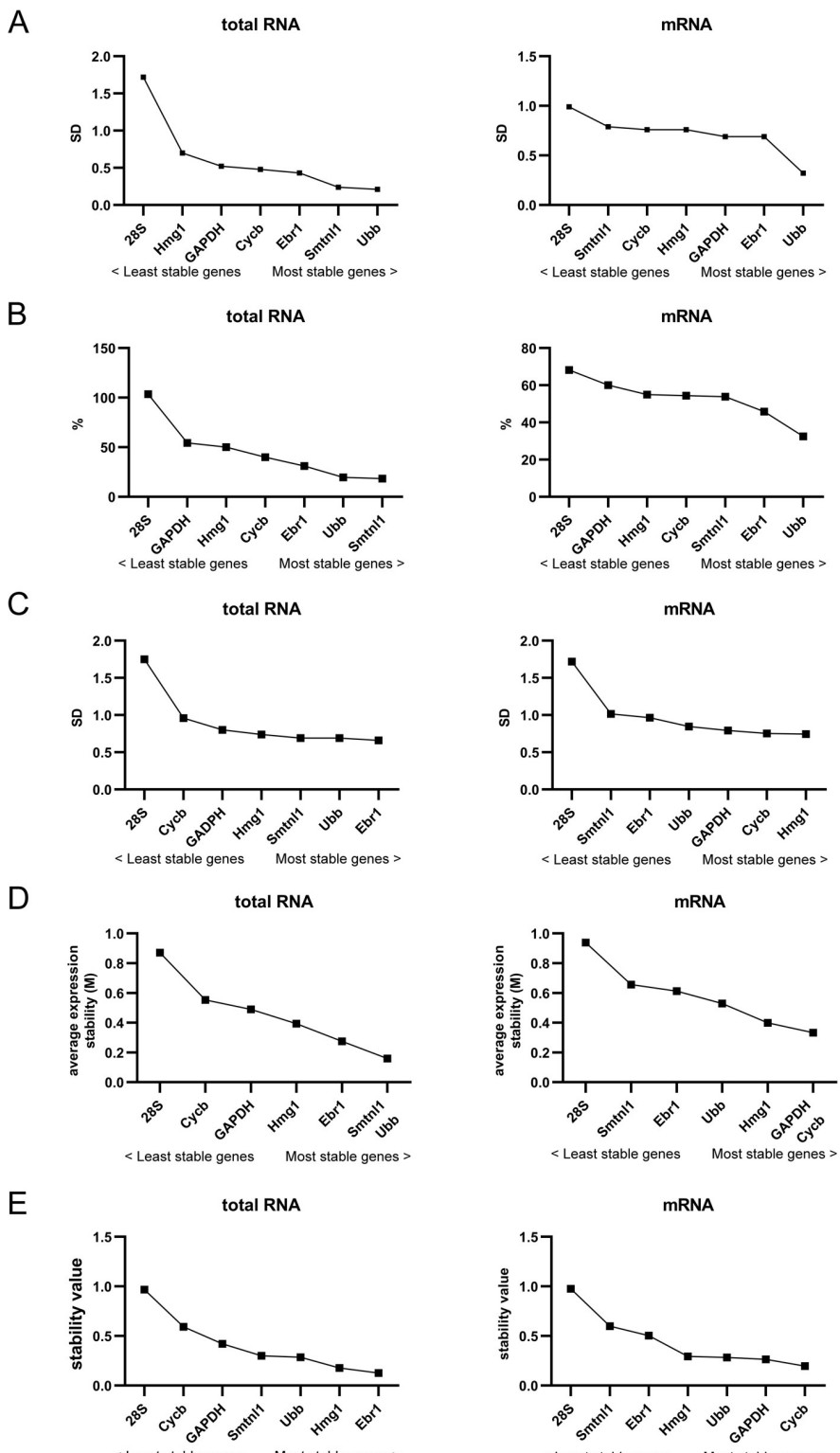

**Fig 3. Stability analysis of candidate reference genes performed by different methods.** Stability estimated by BestKeeper (A), CV (B) ΔCt (C), geNorm (D) and NormFinder (E).

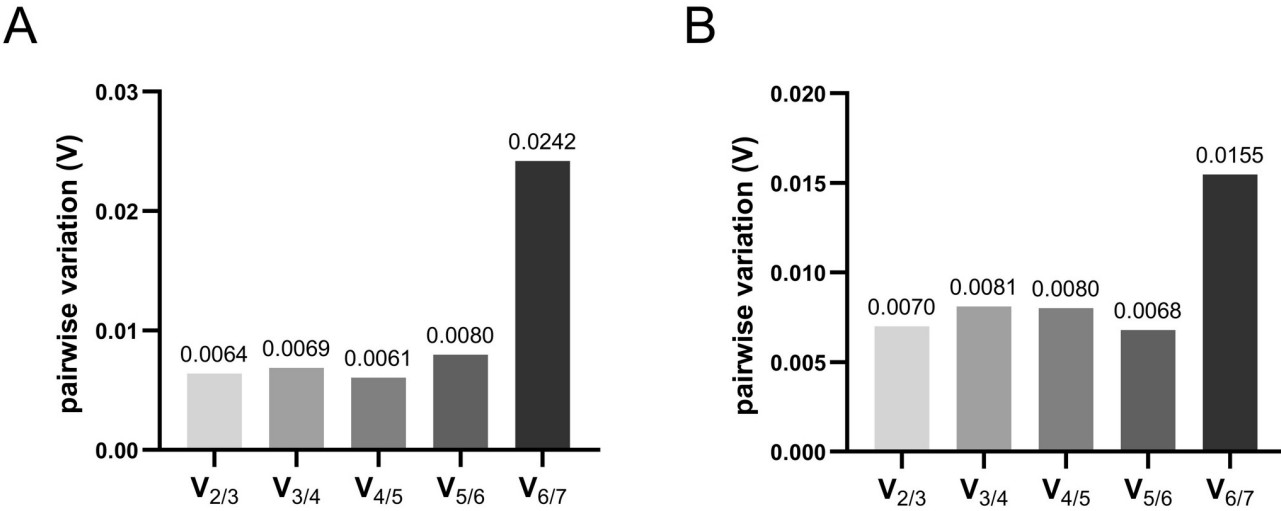

**Fig 4. Determination of minimal number of reference genes for accurate normalization.** Pairwise variations ($V_{n/n+1}$) were calculated by geNorm in all samples (eggs and isolated egg cortices) for total RNA (A) and mRNA (B).

RNA and mRNA. 28S ranked as the least stable gene during analyses of both type of RNA samples.

## Validation of candidate reference genes

To validate the reliability of recommended reference genes, *Daglb-2* was selected from the list of cortically-enriched transcripts. Normalization of signal intensity was done using top-ranked genes and least stable 28S. *Ebr1*, *Smtnl1* and *Ubb* were found to be most appropriate reference genes for total RNA samples (Table 3). After normalization to *Ebr1*, *Smtnl1*, *Ubb* and pair *Smtnl1/Ubb* reference genes, the levels of *Daglb-2* in egg cortices were found to be mildly lower than in eggs (Fig 5A), but these differences between samples and reference genes were statistically insignificant. Normalization to the least stable 28S gene showed 5.64-fold higher level of *Daglb-2* in cortices. In total RNA samples, we could not confirm that *Daglb-2* is cortically-enriched transcript. Nevertheless, nearly equal *Daglb-2* signals normalized to the top-ranked genes in cortices indicate the reliability of chosen reference genes. *Daglb-2* levels

**Table 3. Stability ranking of candidate reference genes given upon BestKeeper, CV, ΔCt, geNorm and NormFinder.**

| Ranking | 1 | 2 | 3 | 4 | 5 | 6 | 7 |
|---|---|---|---|---|---|---|---|
| *total RNA* | | | | | | | |
| **BestKeeper** | *Ubb* | *Smtnl1* | *Ebr1* | *Cycb* | *GAPDH* | *Hmg1* | 28S |
| **CV** | *Smtnl1* | *Ubb* | *Ebr1* | *Cycb* | *Hmg1* | *GAPDH* | 28S |
| **ΔCt** | *Ebr1* | *Ubb* | *Smtnl1* | *Hmg1* | *GAPDH* | *Cycb* | 28S |
| **geNorm** | *Smtnl1/Ubb* | | *Ebr1`* | *Hmg1* | *GAPDH* | *Cycb* | 28S |
| **NormFinder** | *Ebr1* | *Hmg1* | *Ubb* | *Smtnl1* | *GAPDH* | *Cycb* | 28S |
| *mRNA* | | | | | | | |
| **BestKeeper** | *Ubb* | *Ebr1* | *GAPDH* | *Hmg1* | *Cycb* | *Smtnl1* | 28S |
| **CV** | *Ubb* | *Ebr1* | *Smtnl1* | *Cycb* | *Hmg1* | *GAPDH* | 28S |
| **ΔCt** | *Hmg1* | *Cycb* | *GAPDH* | *Ubb* | *Ebr1* | *Smtnl1* | 28S |
| **geNorm** | *Cycb/GAPDH* | | *Hmg1* | *Ubb* | *Ebr1* | *Smtnl1* | 28S |
| **NormFinder** | *Cycb* | *GAPDH* | *Ubb* | *Hmg1* | *Ebr1* | *Smtnl1* | 28S |

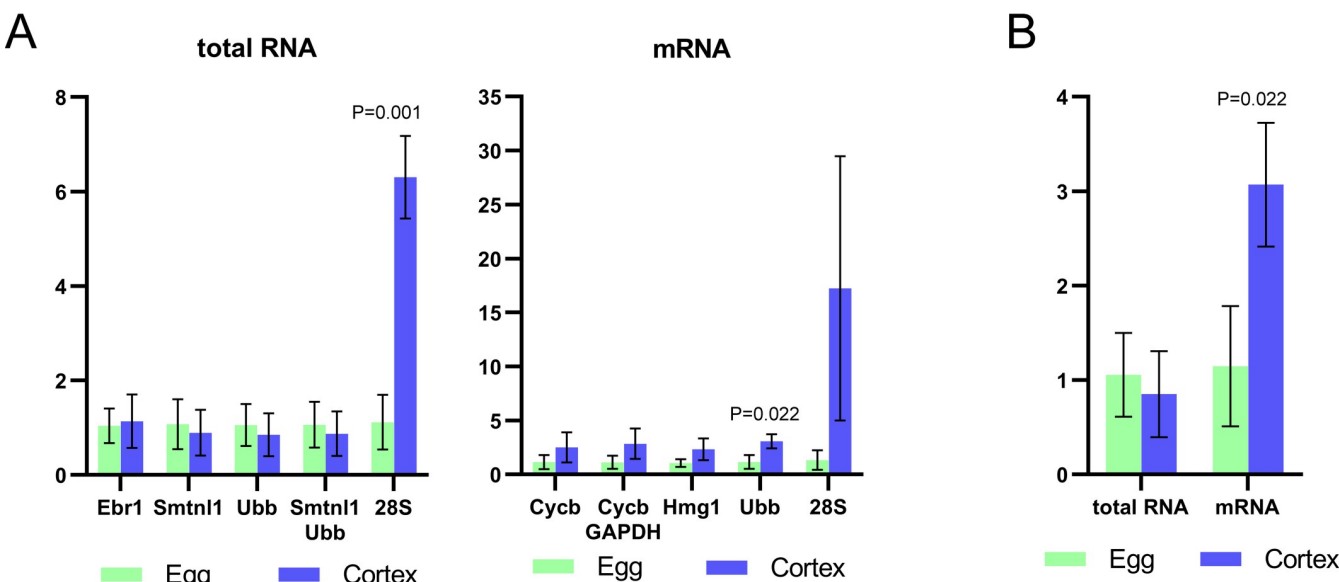

**Fig 5. Comparison of the normalized relative levels of *Daglb-2* between eggs and isolated cortices.** (A) To normalize values, three most stable genes and best pair of genes were selected. Also, data were normalized to least stable gene (28S). (B) Relative levels of *Daglb-2* normalized to *Ubb*. P-values indicate statistical significance between columns based on t-test.

measured in mRNA samples were normalized to appropriate top-ranked genes, *Cycb*, *Hmg1*, *Ubb* and *Cycb*/*GAPDH* (Table 3). In contrast to values calculated in total RNA samples, mRNA analysis showed increased levels of *Daglb-2* in cortices (Fig 5A), from 2.18-fold higher level in case of *Cycb* to 2.65-fold higher level in case of *Ubb*. Although values normalized to *Cycb*, *Hmg1* and *Cycb*/*GAPDH* in cortices were higher than in eggs, they did not reveal statistical significance, which indicate variable Ct values of these genes (Fig 2B). Only values normalized to *Ubb*, with Ct in narrow range, were statistically significant (Fig 5A). We compared relative levels of *Daglb-2* normalized to *Ubb* in total RNA and mRNA (Fig 5B). Total RNA samples did not detect significant difference in mRNA levels between eggs and isolated cortices, while purified mRNA allowed detection of significant enrichment of *Daglb-2* in cortices. This finding suggested that analysis of mRNA is more sensitive than total RNA. 28S which is the worst reference gene in both types of RNA samples showed high *Daglb-2* levels in cortices due to its low levels in cortices.

## Discussion

RT-qPCR is a convenient method to estimate expression of genes in different biological contexts. The primarily goal of this study is to design an approach based on RT-qPCR for quantitative analysis of cortically-associated maternal transcripts of sea urchin eggs. The proposed approach is based on comparison the levels of genes of interest between eggs and isolated egg cortices. The first necessary prerequisite to perform this analysis is a suitable method for RNA isolation. We adapted column-based RNA isolation protocol for isolated egg cortices. Isolated total RNA may be subsequently processed to obtain purified poly(A) RNA. Second prerequisite for accurate RT-qPCR analysis is usage of appropriate reference transcripts (genes), the levels of which are less variable among eggs and isolated cortices. We analyzed four candidate reference genes selected from known reference genes and three relatively abundant transcripts detected in both entire eggs of *S. intermedius* and their isolated cortices. To evaluate comprehensively the stability of selected genes, 28S, *GAPDH*, *Hmg1*, *Ubb*, *Cycb*, *Ebr1* and *Smtnl1*, we

used five different methods and compared the derived results. During analysis of total RNA samples, three programs, BestKeper, ΔCt and geNorm, showed similar results. According to these programs, *Ebr1*, *Smtnl1* and *Ubb* were the three most stable genes (Table 3). Each program ranked these genes differently, but in each case, these genes ranked top 3 as most stable. Results of NormFinder were different, giving *Ebr1*, *Hmg1* and *Ubb* as most stable genes. We decided to designate *Ebr1*, *Ubb* and *Smtnl1* as the most suitable reference genes upon analysis by BestKeper, ΔCt and geNorm. Estimation of level of stability in mRNA samples ranked candidate reference genes differently than in total RNA samples. According to BestKeeper and CV methods *Ubb* was the most stable gene, while other methods showed *Hmg1* (ΔCt), *Cycb* (NormFinder) and *Cycb*/*GAPDH* pair (geNorm) as best reference genes.

Among all the tested genes, only *Ubb* showed suitability for quantitative RNA analysis of total RNA and mRNA samples from isolated egg cortices. *Ubb*, which encodes polyubiquitin, is a well-known reference gene in quantitative expression analysis of sea urchin embryos, as level of *Ubb* mRNA is relatively stable during sea urchin development [39,45]. Also, *Ubb* is suitable for both qualitative and quantitative expression analysis during sea urchin gametogenesis [46,47]. Two new candidate reference genes found by our transcriptomic analysis with abundant transcripts in both eggs and egg cortices, *Ebr1* and *Smtnl1*, previously have not been used as internal control. These two genes were shown to be suitable only for total RNA. *Ebr1* encodes egg cell-surface protein. It is one of proteins that are responsible for species-specific sperm adhesion to sea urchin eggs via interaction with Bindin localized on spermatozoan surface [48,49]. *Smtnl1* encodes a muscle protein that participates in regulation of muscle contraction and adaptation in mammals [50], while in sea urchin embryos its functions remain unstudied.

A main goal of this study was testing the reliability of quantitative approach to analyze transcripts that anchored in subcortical area of eggs. To test the reliability of our approach, we evaluated the expression of any gene of interest that could potentially be associated with egg cortex. Unfortunately, we excluded from our analysis the transcripts have been found to be presumably localized in cortex, that is, *Panda* and *Coup-TF* [14,17]. We have not found *Panda* homolog in *S. intermedius* egg transcriptome. Assembled part of *Coup-TF* is a GC-rich region, which was poorly amplified by RT-qPCR. To find cortex-associated transcript for our analysis, we selected transcripts in transcriptome, the levels (in FPKM) of which were higher in cortices than in eggs. *Daglb-2* was chosen as a transcript that may be localized in egg cortex according to transcriptomic analysis. *Daglb-2* encodes the homolog of mammal transmembrane enzyme diacylglycerol lipase beta. Diacylglycerol lipases alpha and beta localized in plasma membrane generate endocannabinoids from membrane lipids, primarily 2-arachidonoylglycerol. Endocannabinoids are ligands of cannabinoid receptors. Binding endocannabinoids with receptors activate inflammatory response in macrophages and neural cells and inhibit the release of neurotransmitters in central and peripheral nervous systems [51–53]. Endocanabinoids also play a significant role in many reproductive events of invertebrates and vertebrates [54]. Experiments that showed inhibition of acrosomal reaction by either synthetic or natural cannabinoids from marihuana suggest the presence of cannabinoid receptors on sea urchin spermatozoan surface and their significance in polyspermy blockage [55,56]. Later on, transcriptome analysis confirmed the presence of mRNA sequence of cannabinoid receptor 1 in sea urchin testes [57]. Sea urchin ovaries contain endocannabinoid, anandamide, which is other common ligand for cannabinoid receptors [58]. Although the presence of 2-arachidonoylglycerol have not studied in sea urchin eggs, Daglb-2 may be necessary for the synthesis of 2-arachidonoylglycerol, which is probably required to prevent polyspermy.

In unfertilized eggs, cortical localization of *Daglb-2* may be necessary for local translation of the encoded putative transmembrane protein, which is integrated in plasma membrane.

Although translation is significantly increased after fertilization, slow rate of protein synthesis has been found in unfertilized eggs [59,60]. Eggs can be stored in ovaries for weeks to months until spawning [61] and translation may require maintaining metabolism via renewed protein pool during long-term egg storage. Local translation probably takes place in mitotic spindles of early sea urchin embryos [62], but is unknown for cortical regions of sea urchin eggs and early embryos. Commonly, transmembrane proteins are translated on ribosomes associated with rough endoplasmic reticulum. Transmembrane proteins are introduced and subsequently folded in endoplasmic reticulum membrane. Plasma membrane proteins are translocated to Golgi apparatus and then to cell membrane. The presence of rough endoplasmic reticulum and Golgi bodies in cortical region of sea urchin eggs [19,63–66] suggest that transmembrane Daglb-2 may be translated in cortical area. Taking into account our transcriptomic analysis and predicted functions of *Daglb-2*, we propose this gene to be suitable as a gene of interest to approbate approach of quantitative evaluation of cortex-associated transcripts in sea urchin eggs.

Many researchers normalize signal intensities of studied genes against a single reference gene. Nevertheless, it is necessary to confirm invariant expression of potential reference gene under all experimental conditions. Alternatively, usage of two or more reference genes is the better choice [27,67]. geNorm allows determination of number of reference genes for reliable normalization (two genes is minimal number). Our analysis did not show significant difference among one type of samples (total RNA or mRNA). Fundamental differences were found between total RNA and mRNA data. Total RNA samples did not show significant differences of *Daglb-2* levels between eggs and their cortices. Analysis of purified mRNA revealed increased *Daglb-2* levels in cortices supporting our differential expression results. While normalization to any suitable reference gene showed similar ratios of *Daglb-2* levels between eggs and cortices, only using *Ubb* exhibited significant differences. Taking into account these results and lowest Ct variation of *Ubb* among evaluated genes, we propose that *Ubb* is the best choice for intensity signal normalization. *Ubb* was defined as the most stable gene by Best Keeper and CV methods. Different methods for evaluation of expression stability are based on different principles. They may well define gene as a least stable, but the most stable genes are different [68]. In case of total RNA samples, we propose to use any of the following genes: *Ubb*, *Ebr1* and *Smtnl*.

An important question is why *Daglb-2* cortical enrichment is detected only in case of poly (A) RNA analysis. Increased levels of *Daglb-2* in isolated cortexes were detected by mRNA transcriptome analysis and also by RT-qPCR using poly(A)-enriched samples, while RT-qPCR of total RNA samples showed similar levels of *Daglb-2* in both eggs and cortices. Evidently, a major cause of different ratios of *Daglb-2* levels between total and mRNA templates is different amounts of polyadenylated RNAs in eggs and cortices. According to our data, percentage of mRNA in cortex-associated RNA pool is 0.64%, which 2.27-fold lower than in entire eggs. Hence, total RNA templates from eggs and cortices are markedly differ by mRNA amounts, which led to misrepresentation of *Daglb-2* levels. Also, analysis of poly(A) RNA offers additional advantage in comparison to total RNA. It is known that usage of mRNA templates may result in greater sensitivity of RT-qPCR, which improve measurements of transcript levels [69,70]. Lower sensitivity of RT-qPCR with total RNA may also affect quantification of *Daglb-2* levels in eggs and cortices.

## Conclusions

This study provided the first report on efficacy of RT-qPCR analysis of maternal transcript to verify its association with egg cortex in sea urchin eggs. Firstly, optimization of RNA isolation

method using isolated cortices indicated feasibility of extracting adequate levels of RNA required for cDNA synthesis. Next, evaluation of possible reference genes allowed the determination of appropriate genes that can be used for signal intensity normalization either for total RNA or mRNA samples. Finally, RT-qPCR analysis of presumably cortex-associated *Daglb-2* showed increased levels of its transcript in egg cortices revealing its correlation with transcriptomic analysis. Thereby, RT-qPCR may be utilized as one of methods to verify cortex association of mRNAs in sea urchin eggs using poly(A) RNA templates.

## Supporting information

**S1 Fig. Melt curve analysis of PCR products from one biological replicate.** Egg and cortex melt curves are given together for each gene. Negative control (template-free) is marked by Neg.
(TIF)

**S1 Table. Raw Ct values, calculated means and standard deviations in all samples.**
(XLSX)

## Acknowledgments

The authors are grateful to Dr. Andrey Kukhlevsky for Sanger sequencing. Confocal images were performed using equipment of the Joint-Use Center "Biotechnology & Genetic Engineering" of FSCEATB FEB RAS. This research was supported in through computational resources provided by the Shared Services Center "Data Center of FEB RAS".

## Author Contributions

**Conceptualization:** Konstantin V. Yakovlev.

**Data curation:** Yulia O. Kipryushina, Konstantin V. Yakovlev.

**Formal analysis:** Mariia A. Maiorova.

**Investigation:** Yulia O. Kipryushina, Mariia A. Maiorova, Konstantin V. Yakovlev.

**Methodology:** Mariia A. Maiorova, Konstantin V. Yakovlev.

**Supervision:** Konstantin V. Yakovlev.

**Validation:** Yulia O. Kipryushina.

**Writing – original draft:** Yulia O. Kipryushina, Konstantin V. Yakovlev.

**Writing – review & editing:** Yulia O. Kipryushina, Mariia A. Maiorova, Konstantin V. Yakovlev.

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
