## [Decision Letter · Decision Letter 0]

22 Dec 2021

PONE-D-21-36401A method to quantitate maternal transcripts localized in sea urchin egg cortex by RT-qPCR with accurate normalizationPLOS ONE

Dear Dr. Yakovlev,

Thank you for submitting your manuscript to PLOS ONE. After careful consideration, we feel that it has merit but does not fully meet PLOS ONE’s publication criteria as it currently stands. Therefore, we invite you to submit a revised version of the manuscript that addresses the points raised during the review process.

We look forward to receiving your revised manuscript.

Kind regards,

Wei Cui, Ph.D.

Academic Editor

PLOS ONE

Journal Requirements:

[This work was supported by the Russian Foundation for Basic Research (Grant number: 20-04-00332). The authors are grateful to Dr. Andrey Kukhlevsky for Sanger sequencing.]

 [NO]

Additional Editor Comments:

Both expert reviewers found this manuscript interesting. However, several major questions were also pointed out and authors need to address them in their revision. Please refer to the reviewers' comments for details.

Reviewers' comments:

Reviewer's Responses to Questions

**Comments to the Author**

1. Is the manuscript technically sound, and do the data support the conclusions?

Reviewer #1: Partly

Reviewer #2: Partly

2. Has the statistical analysis been performed appropriately and rigorously? 

Reviewer #1: I Don't Know

Reviewer #2: N/A

3. Have the authors made all data underlying the findings in their manuscript fully available?

Reviewer #1: Yes

Reviewer #2: Yes

4. Is the manuscript presented in an intelligible fashion and written in standard English?

Reviewer #1: No

Reviewer #2: Yes

5. Review Comments to the Author

Reviewer #1: This study attempts to use RT-qPCR as a means of quantitating maternal mRNA transcripts in the cortex of sea urchin eggs. The paper concentrates on the evaluation of seven potential reference genes that are ranked based on the results of five different methods and then indicates that the gene of interest Daglb-2 shows increased levels of mRNA in cortices relative to the best reference genes. I have the following comments about this research.

Major Points:

1) Unlike the case with Xenopus, C. elegans or Drosophila eggs, very few polarized maternal proteins or mRNAs have been identified in sea urchin eggs. This despite extensive research effort over several decades. The sea urchin early embryo is often cited as a prime example of a highly regulative embryo with irreversible polarization setting up only once the embryo has reached the 16 cell stage. The authors suggest that RT-qPCR would be useful for the identification of maternal mRNAs that are potentially enriched in the cortex. They indicate that the mRNA for the genes Panda (ref 14) and Coup-TF (refs 16, 17) are associated with subcortical areas of oocytes, eggs, and embryos. However, although in situ hybridization (ISH) imaging does suggest a gradient of Panda in oocytes, cortical association is not clear and even the Panda gradient itself is not obvious in eggs (Fig 5 in ref 14). In addition, the apparent cortical association of Coup-TF in the oocyte/egg appears to be present in only two of three sea urchin species tested (refs 16, 17). Given that these mRNAs have some literature precedence for cortical association it would make sense to use then as targets for the RT-qPCR experiment that the authors conduct in the present study. However, the authors indicate that they excluded these transcripts because they were unable to identify a Panda homologue in their species and that the G-C rich nature of Coup-TF made it difficult to amplify via RT-qPCR. Instead they choose to focus on the transcript of the gene Daglb-2 which they indicate was identified by transcriptomic analysis. The authors need to provide additional details about the nature of how this transcriptomic analysis was performed and what were the results. For example, were cortices isolated in a similar manner? Did other transcripts show cortical enrichment? Also, a biological role for the suggested cortical localization of the Daglb-2 protein is not provided. Do the authors have a hypothetical rationale why endocannabinoid signaling would be subject to polarized maternal distribution? They suggest that the transcript is localized in the cortex because of a need for local translation and integration into the membrane. Are the authors aware of precedence in the sea urchin for the localized cortical translation of membrane proteins? In general, the authors need to provide a better argument about why they ended up focusing on Daglb-2.

2) Given that this is a “proof-of-concept” study it is important that the authors provide the appropriate documentation of their methods. The description that they give in the Materials and Methods section of the cortex isolation process (page 5-6) is inadequate and incomplete. In addition, they need to provide low and high magnification phase contrast microscopy images of the cortical preparations that they generate for RNA isolation. These images would make it clear how intact the cortices were, to include the presence or absence of cortical granules.

3) This study would be greatly improved by providing RNA FISH localization evidence of the cortical association of Daglb-2. This evidence would help validate the results of their RT-qPCR results. The senior author has previously published work involving confocal microscopy of the immunofluorescent localization of proteins in sea urchin egg cortices (ref 19) so the expertise and imaging instrumentation is available. In addition, it would be important to provide FISH and/or ISH images of Daglb-2 in whole oocytes, eggs and early embryos.

Minor Points:

1) The paper needs to be carefully edited for proper English grammar, word choice and sentence structure. In its current form, it is sometimes difficult to decipher the meaning of what is written.

2) What type of statistical analysis was performed on the data in Fig 4? T tests? ANOVA?

3) The fact that Daglb-2 transcripts were found to be cortex-enriched only in mRNA and not in total RNA analysis is puzzling. The authors should speculate on what might be causing this difference.

Reviewer #2: In this manuscript, authors tried to establish a method for quantitate maternal genes in eggs of a sea urchin, Strongylocentrotus intermedius. Especially, authors tried to find the stable reference gene, which is essential for the qPCR analysis. Personally, I recognize this interesting, but I also have concerns.

Obtaining RNA and the usage of template RNA for cDNA synthesis look not appropriate. For example, authors showed Ct in Fig1, in which authors treat the data as absolute quantity. However, authors obtained RNA from countless eggs and set RNA to 1 µg to synthesize cDNA. If the authors need the absolute quantity, the authors need to count the number of eggs, to use one/two step RT-qPCR kit, by which we do not have to change tubes, and same egg-number volume of cDNA should be applied into each qPCR tube. In another example, the protocol of cortex isolation is not stable (although the authors described it stable). This is because nobody can expect how much % of egg cortex are remained on the coverslip. In addition, RNA composition might be different between each pole of the egg since the protein localization is different between animal and vegetal poles. I also have a question how much maternal mRNA is trapped by poly(A) selection. It is well known that poly-adenylation system starts at the fertilization, meaning that un-fertilized eggs contain a number of non-poly(A) RNA. So, it is better for authors to explain such RNA characteristics (if I were wrong, please ignore this point).

Therefore, if the authors intend to make “standard” methods for maternal RT-qPCR for eggs/cortex, initially it is necessary to set the protocol much more precisely in each step.

6. PLOS authors have the option to publish the peer review history of their article (what does this mean?). If published, this will include your full peer review and any attached files.

Reviewer #1: No

Reviewer #2: No

---

## [Author Response · Author response to Decision Letter 0]

6 Apr 2022

Comments concerning the Journal Requirements:

"When submitting your revision, we need you to address these additional requirements.

https://journals.plos.org/plosone/s/file?id=ba62/PLOSOne_formatting_sample_title_authors_affiliations.pdf."

- We re-formatted the Manuscript according to the PLOS ONE’s requirements.

"2. We note that the grant information you provided in the ‘Funding Information’ and ‘Financial Disclosure’ sections do not match. 

[This work was supported by the Russian Foundation for Basic Research (Grant number: 20-04-00332). The authors are grateful to Dr. Andrey Kukhlevsky for Sanger sequencing.]

Please remove any funding-related text from the manuscript and let us know how you would like to update your Funding Statement. Currently, your Funding Statement reads as follows: [NO]

Please include your amended statements within your cover letter; we will change the online submission form on your behalf."

- We removed funding information in Acknowledgments and provide a grant number in Funding Information section and in the Cover Letter.

"4. We note that you have indicated that data from this study are available upon request. PLOS only allows data to be available upon request if there are legal or ethical restrictions on sharing data publicly. For more information on unacceptable data access restrictions, please see http://journals.plos.org/plosone/s/data-availability#loc-unacceptable-data-access-restrictions. 

We will update your Data Availability statement on your behalf to reflect the information you provide."

- We have no restrictions to provide our data. We uploaded all required data in GenBank (SRAs and assembled transcriptome are available in BioProject PRJNA686841).

"5. Please include your full ethics statement in the ‘Methods’ section of your manuscript file. In your statement, please include the full name of the IRB or ethics committee who approved or waived your study, as well as whether or not you obtained informed written or verbal consent. If consent was waived for your study, please include this information in your statement as well."

- In our case, the type of ethics statement is Field Research. We added number of Permission and name of agency in Materials and Methods. Also, this information is given in online submission form.

"6. Please include captions for your Supporting Information files at the end of your manuscript, and update any in-text citations to match accordingly. Please see our Supporting Information guidelines for more information: http://journals.plos.org/plosone/s/supporting-information."

- We fitted Supporting Information captions to the PLOS ONE’s requirements and included these captions at the end of the manuscript. 

Additional Editor Comments:

Both expert reviewers found this manuscript interesting. However, several major questions were also pointed out and authors need to address them in their revision. Please refer to the reviewers' comments for details.

Answers to the reviewers’ comments

Reviewer #1: This study attempts to use RT-qPCR as a means of quantitating maternal mRNA transcripts in the cortex of sea urchin eggs. The paper concentrates on the evaluation of seven potential reference genes that are ranked based on the results of five different methods and then indicates that the gene of interest Daglb-2 shows increased levels of mRNA in cortices relative to the best reference genes. I have the following comments about this research.

Major Points:

"1) Unlike the case with Xenopus, C. elegans or Drosophila eggs, very few polarized maternal proteins or mRNAs have been identified in sea urchin eggs. This despite extensive research effort over several decades. The sea urchin early embryo is often cited as a prime example of a highly regulative embryo with irreversible polarization setting up only once the embryo has reached the 16 cell stage."

- Indeed, sea urchin early development is often referred as regulative. Though, this point of view is not common and debate is still going on. We believe that both induction and determination (or reversive specification) take place in early sea urchin embryos. It suggests that several localized maternal factors should be involved in development regulation.

"The authors suggest that RT-qPCR would be useful for the identification of maternal mRNAs that are potentially enriched in the cortex. They indicate that the mRNA for the genes Panda (ref 14) and Coup-TF (refs 16, 17) are associated with subcortical areas of oocytes, eggs, and embryos. However, although in situ hybridization (ISH) imaging does suggest a gradient of Panda in oocytes, cortical association is not clear and even the Panda gradient itself is not obvious in eggs (Fig 5 in ref 14). In addition, the apparent cortical association of Coup-TF in the oocyte/egg appears to be present in only two of three sea urchin species tested (refs 16, 17). Given that these mRNAs have some literature precedence for cortical association it would make sense to use then as targets for the RT-qPCR experiment that the authors conduct in the present study. However, the authors indicate that they excluded these transcripts because they were unable to identify a Panda homologue in their species and that the G-C rich nature of Coup-TF made it difficult to amplify via RT-qPCR. Instead they choose to focus on the transcript of the gene Daglb-2 which they indicate was identified by transcriptomic analysis.

The authors need to provide additional details about the nature of how this transcriptomic analysis was performed and what were the results. For example, were cortices isolated in a similar manner? Did other transcripts show cortical enrichment?"

- Cortex preparation for transcriptomic analysis and subsequent RNA isolation were done as described in Materials and Methods. Now, we provide more detailed description of transcriptomic analysis. SRAs and assembled transcriptome were submitted to GenBank. We give only a part of our results concerning expression of reference genes and Dagbl-2 (Table 1). We found a number of cortex-associated transcripts and will prepare separate paper, which will include full transcriptomic data. In the manuscript, we present approach, which allow to find out cortical association of transcript by RT-qPCR. This approach we will use to verify cortical association of mRNAs discovered by transcriptomic analysis.

"Also, a biological role for the suggested cortical localization of the Daglb-2 protein is not provided. Do the authors have a hypothetical rationale why endocannabinoid signaling would be subject to polarized maternal distribution? They suggest that the transcript is localized in the cortex because of a need for local translation and integration into the membrane. Are the authors aware of precedence in the sea urchin for the localized cortical translation of membrane proteins? In general, the authors need to provide a better argument about why they ended up focusing on Daglb-2."

- In Discussion, we give explanation of biological role of Daglb-2 in plasma membrane of unfertilized eggs. We also cite data that point to the existence of local translation in sea urchin egg cortex.

"2) Given that this is a “proof-of-concept” study it is important that the authors provide the appropriate documentation of their methods. The description that they give in the Materials and Methods section of the cortex isolation process (page 5-6) is inadequate and incomplete."

- At first, we decided do not give description of cortex isolation procedure in details, but now we provide all necessary information about this method. Three years ago, Henson and co-authors published methodological article about cortex isolation with following manipulation for microscopy [31, Henson JH, Samasa B, Burg EC. High resolution imaging of the cortex isolated from sea urchin eggs and embryos. Methods Cell Biol. 2019;151:419-32], which we missed to cite. We think that this article is excellent resource to study this method.

"In addition, they need to provide low and high magnification phase contrast microscopy images of the cortical preparations that they generate for RNA isolation. These images would make it clear how intact the cortices were, to include the presence or absence of cortical granules."

- Sure, for RNA isolation isolated cortices should be intact. We always do visual observation of coverslips with isolated cortices to check remaining undisrupted eggs. If necessary, we observe cortex samples by phase contrast at low magnifications (Fig. 1A). Integrity of cortical layers we verify by using DIC and confocal imaging at higher magnifications (Fig. 1B, C). We do not it each experiment because we found that underdisrupted eggs and overdisrupted cortices are visible at low magnifications. In the revised version, we give DIC and confocal images to confirm a good quality of cortical samples by the presence of cortical granules and pattern of actin staining. Description of sample preparation for microscopy we give in Materials and Methods section and results of our observation we added to Results (Quality of isolated cortices and measurement of RNA amount).

"3) This study would be greatly improved by providing RNA FISH localization evidence of the cortical association of Daglb-2. This evidence would help validate the results of their RT-qPCR results. The senior author has previously published work involving confocal microscopy of the immunofluorescent localization of proteins in sea urchin egg cortices (ref 19) so the expertise and imaging instrumentation is available. In addition, it would be important to provide FISH and/or ISH images of Daglb-2 in whole oocytes, eggs and early embryos."

- Study of RNA localization by in situ hybridization will be direct evidence of Daglb-2 cortical localization. In present work, we primarily focus on approach for detection of cortical association by RT-qPCR. Before RT-qPCR cortical association of Daglb-2 was found by transcriptomic analysis. We think that combination of these two methods is satisfactory to claim about cortical localization of Daglb-2 mRNA. We are planning to get data about cortical localization of Daglb-2 and other transcripts by in situ hybridization, which will be submitted as other paper.

Minor Points:

"1) The paper needs to be carefully edited for proper English grammar, word choice and sentence structure. In its current form, it is sometimes difficult to decipher the meaning of what is written."

- To fix grammatical and spelling errors, we used “ManuscriptEdit” editing service.

"2) What type of statistical analysis was performed on the data in Fig 4? T tests? ANOVA?"

- We determined a statistical significance by t-test between pairs of columns in each type of sample (total RNA and mRNA). Now, we give statistically significant P values above the columns in Fig. 5.

"3) The fact that Daglb-2 transcripts were found to be cortex-enriched only in mRNA and not in total RNA analysis is puzzling. The authors should speculate on what might be causing this difference."

- In the first version of the manuscript, we omitted speculations about this issue. Now, we concern this point in Discussion.

Reviewer #2: In this manuscript, authors tried to establish a method for quantitate maternal genes in eggs of a sea urchin, Strongylocentrotus intermedius. Especially, authors tried to find the stable reference gene, which is essential for the qPCR analysis. Personally, I recognize this interesting, but I also have concerns.

"Obtaining RNA and the usage of template RNA for cDNA synthesis look not appropriate. For example, authors showed Ct in Fig1, in which authors treat the data as absolute quantity. However, authors obtained RNA from countless eggs and set RNA to 1 µg to synthesize cDNA. If the authors need the absolute quantity, the authors need to count the number of eggs, to use one/two step RT-qPCR kit, by which we do not have to change tubes, and same egg-number volume of cDNA should be applied into each qPCR tube."

- The data represented in Figure 2 (Fig. 1 in previous version) illustrate a dispersion of Ct values among all samples for each reference gene and require for further gene expression stability analysis. This is a first stage for evaluation of gene stability. Genes that reveal narrow range of Ct values should be ranked as most appropriate reference genes in subsequent analysis using different methods (GeNorm, BestKeeper etc).

"In another example, the protocol of cortex isolation is not stable (although the authors described it stable). This is because nobody can expect how much % of egg cortex are remained on the coverslip. In addition, RNA composition might be different between each pole of the egg since the protein localization is different between animal and vegetal poles."

- We believe that the proposed approach cannot be considered as a tool for absolute measurements for the reasons noticed by the reviewer. Peng and Wikramanayake published article [13, Peng CJ, & Wikramanayake AH. Differential regulation of disheveled in a novel vegetal cortical domain in sea urchin eggs and embryos: implications for the localized activation of canonical Wnt signaling. PloS one. 2013; 8: e80693], where they provided localization of Disheveled in isolated cortices (Fig. 2), this protein is not present on all cortexes. It means that parts of the cortices that does not bind to the glass are removed during their isolation. This also means that not all cortexes will contain transcripts that can be localized to the animal or vegetative poles. The percentage of remaining egg cortex may also vary. For this reason, the cortex isolation method would be inappropriate in terms of determining absolute expression levels and counting how many of the studied transcripts are present in whole eggs and their isolated cortexes. On the other hand, the remaining part of the cortical layer of the oocytes on the slides remains intact and contains localized transcripts. Based on this, we believe that this cortex extraction protocol is unstable in terms of absolute calculations of the number of transcripts. Nevertheless, our data show that cortex isolation can be successfully used to obtain RNA followed by comparison of the relative levels of transcripts in oocytes and their isolated cortexes, as demonstrated in the manuscript. We suggest that our approach can be one of the ways for the association of the studied transcripts with the cortical layer, hence their cortical localization. We believe that eggs can bind to coverslip with different sides equally, which means that RNA samples will contain both vegetal and animal transcripts, if any.

"I also have a question how much maternal mRNA is trapped by poly(A) selection. It is well known that poly-adenylation system starts at the fertilization, meaning that un-fertilized eggs contain a number of non-poly(A) RNA. So, it is better for authors to explain such RNA characteristics (if I were wrong, please ignore this point)."

- This is reasonable question and we explain below why poly(A) selection should be valid for unfertilized egg analysis.

This is the case that rate of cytoplasmic polyadenylation increases after fertilization of sea urchin embryos. According to measurements of Wilt (Wilt FH. The dynamics of maternal poly (A)-containing mRNA in fertilized sea urchin eggs. Cell. 1977; 11: 673-681), average length of poly(A) tails was 45 nucleotides for unfertilized eggs and 60 nucleotides for zygotes. Possibly, some maternal transcripts can be presented in both polyadenylated and deadenylated forms in unfertilized sea urchin eggs that may misrepresent quantitative measurements after poly(A) selection. Though, usual mechanism for mRNA storage and masking is partial deadenylation of poly(A) tails to 20-40 nucleotides, which is known for mice, worms and some other animals. Taking into account these data, we think that poly(A) selection is suitable for analysis of most transcripts of sea urchin eggs. A one of successful examples of poly(A) RNA selection from eggs for NGS means that most maternal RNAs are not fully deadenylated (Tu Q, Cameron RA, Worley KC, Gibbs RA, & Davidson, EH. Gene structure in the sea urchin Strongylocentrotus purpuratus based on transcriptome analysis. Genome Research. 2012; 22: 2079-2087) and their poly(A) tails can bind to oligo(d)T primers. 

The next point is about percentage of purified poly(A) RNA. We accounted percentage of poly(A) RNA purified from egg and cortex total RNA samples in each experiment. In egg samples, poly(A) RNA was 1.3-1.7% from total RNA before purification that correspond to approximate rate of mRNA in eukaryotic cells (1-5%). In cortex samples, poly(A) RNA percent was always lower than in eggs (0.5-0.72%). Lower rate of mRNA in cortices may be one of reasons of different ratios of Daglb-2 levels between total RNA and mRNA. We included our calculations in the Results (Quality of isolated cortices and measurement of RNA amount). Discussion of possible reasons of the difference in RT-qPCR data obtained from total RNA and mRNA we give in Discussion.

"Therefore, if the authors intend to make “standard” methods for maternal RT-qPCR for eggs/cortex, initially it is necessary to set the protocol much more precisely in each step."

- We think that we should not give particular description in each step, because our approach consists of set of well-know methods. Detailed description of cortex isolation procedure is presented in resent methodological article [31, Henson JH, Samasa B, Burg EC. High resolution imaging of the cortex isolated from sea urchin eggs and embryos. Methods Cell Biol. 2019;151:419-32]. Indeed, we omitted several key points in “Materials and Methods” that complicate understanding presented data. Now, we give all necessary information sufficient for deep analysis and to reproduce our results. Also, as we noted above, we changed title to be more relevant to the manuscript content.

General comments

We added and re-ordered citations in the text and some ones have been excluded. New and re-ordered cited articles we marked by track changes.

---

## [Editor Report · Decision Letter 1]

14 Apr 2022

An approach to quantitate maternal transcripts localized in sea urchin egg cortex using RT-qPCR with accurate normalization

PONE-D-21-36401R1

Dear Dr. Yakovlev,

We’re pleased to inform you that your manuscript has been judged scientifically suitable for publication and will be formally accepted for publication once it meets all outstanding technical requirements.

Kind regards,

Wei Cui, Ph.D.

Academic Editor

PLOS ONE

Additional Editor Comments (optional):

Questions and concerns have been addressed.